# Uncommon Magnetism in Rare-Earth Intermetallic Compounds with Strong Electronic Correlations

**Pavel S. Savchenkov** [1,2,*] and **Pavel A. Alekseev** [1,2]

1    National Research Centre "Kurchatov Institute", 123182 Moscow, Russia
2    Institute for Laser and Plasma Technologies, National Research Nuclear University "MEPhI", 115409 Moscow, Russia
*    Correspondence: savch92@gmail.com

**Abstract:** Rare-earth intermetallic compounds are characterised by the presence of a long-range magnetic order due to the interaction of local magnetic moments periodically located within the crystal lattice. This paper considers the possibility of forming an ordered state in cases where there is no opportunity to observe the local moment of the f-electronic shell in a traditional sense. These are, first of all, systems with a singlet ground state, as well as systems with fast spin fluctuations caused by a homogeneous intermediate-valence state of a rare-earth ion. Extensive experimental studies of these effects using neutron diffraction, neutron spectroscopy, and high-pressure studies of the magnetic phase diagram are presented and analysed, and the corresponding microscopic model representations are discussed. In particular, the possible origin of long-range magnetic order in mixed-valence compounds is analysed.

**Keywords:** intermediate valence; long-range magnetic order; crystalline electric field; induced magnetism; neutron spectroscopy

## 1. Introduction

Rare-earth intermetallic compounds with strong electronic correlations represent a unique class of materials that is attracting increasing attention from researchers in the field of condensed matter physics. One of the key features of rare-earth intermetallic compounds is the presence of strong electronic correlations. This means that electrons in these systems strongly interact with each other, leading to a variety of unique properties observed in intermetallic systems, such as the formation of intermediate valence (IV) [1] and heavy fermion (HF) states [2], Kondo insulators [3,4], high-temperature superconducting systems [5], colossal magnetoresistance [6], and others, including laser-induced phenomena [7,8].

The magnetism of rare-earth intermetallics is an important topic to discuss. From the perspective of classical metallic compounds, rare-earth intermetallics are considered unusual because they have an uncompensated local magnetic moment associated with electrons in the inner 4f shell. When studying the magnetic properties of these materials, it is important to consider that the interactions between 4f electrons and the nucleus, as well as between the 4f electrons themselves within the f-ion, play a significant role (electrostatic and spin-orbit interactions). These interactions determine the electronic structure of the rare-earth ion. Additionally, the partially filled 4f shell acts as an inner shell, and the external influence on it is shielded by the $5s^2$ and $5p^6$ electron shells. Therefore, the interactions between 4f electrons and the crystalline electric field, as well as the exchange interaction, are relatively weak.

Many unique characteristics of rare-earth magnets are related to the specific nature of exchange interactions involving rare-earth ions. The direct exchange interaction between rare-earth ions in the intermetallic compounds is usually negligible, and the main role is played by the indirect interaction through conduction electrons known as the Ruderman–Kittel–Kasuya–Yosida (RKKY) interaction.

In fact, when we refer to the common magnetism of rare-earth intermetallics, we mean the alignment of localised magnetic moments through indirect RKKY exchange interactions. However, there are rare-earth intermetallic systems in which the magnetism is considered atypical or uncommon. Primarily, these are systems characterised by induced magnetism, which is the phenomenon of forming an ordered magnetic state at low temperatures in systems with a singlet ground state. The phenomenon of induced magnetism was first theoretically predicted for a simple model system consisting of two singlets [9–12], and it was later experimentally observed in a study [13] on the intermetallic compound PrNi.

Another type of compounds in which ordered magnetic states are uncommon is intermetallic compounds containing intermediate-valence rare-earth ions. It is assumed that, in these systems, two electron states of the ion—one localised on the 4f orbital of the atom and the other in the conduction band—are energetically close. Due to strong electron correlations, transitions between different charge and spin configurations become possible. As a result, there is a partial delocalisation of the 4f states, leading to non-integer occupancy of the 4f shell, which has been experimentally observed [14]. Intermediate valence is typically observed in elements from the beginning, middle, and end of the lanthanide series (Ce, Sm, Eu, Tm, and Yb). Intermediate valence, obviously, does not contribute to magnetism: fast spin fluctuations with a characteristic time $10^{-12}$–$10^{-13}$ s introduce disorder into the structure of magnetic moments. Moreover, the overwhelming majority of the previously known intermediate-valence compounds are formed based on Ce and Yb, in which the intermediate valence appears as a result of competition between the magnetic and nonmagnetic configurations ($f^1$–$f^0$ and $f^{13}$–$f^{14}$, respectively), which also, apparently, does not favor of the formation of long-range magnetic order.

The first intermediate-valence system where a transition to a magnetically ordered state was reliably observed is TmSe, with an average valence of $\nu_{Tm} \approx 2.6$ [15,16]. At temperatures below $T_N = 3.45$ K, the system undergoes a transition to an ordered state of the antiferromagnetic (AFM) type. This case is discussed in more detail in Section 3.

Intermediate valence for elements from the middle of the rare-earth series, such as Sm and Eu, bears some resemblance to Ce and Yb. In the intermediate-valence compounds of both Sm and Eu, there exists a nonmagnetic competing configuration $^7F_0$ ($f^6$). In Sm's intermediate-valence compounds, magnetism has not been reliably detected. On the contrary, wide regions of the pressure (substitution)–temperature phase diagram, which correspond both to the ground state with the long-range magnetic order (LRMO) of AFM-type (typical $T_N \sim$10–30 K) and intermediate-valence state of europium, were obtained for Eu-based 1-2-2 compounds ($EuCu_2(Si,Ge)_2$ [17–22], $EuPd_2Si_2$ [23], $EuPt_2Si_2$ [24] $EuNi_2P_2$ [25], $EuNi_2Ge_2$ [26,27], and $EuRh_2Si_2$ [28,29]).

In this review, we delve into the realm of uncommon magnetism in rare-earth intermetallic compounds with strong electronic correlations, i.e., the induced magnetic ordering (Section 2) as well as LRMO in intermediate-valence compounds (Section 3). To achieve this, we have employed the results from the combination of experimental techniques, including thermodynamic as well as neutron spectroscopy methods.

The experience of the performed research in the field of physics of systems with strong electron correlations has shown that methods based on thermal neutron scattering are among the most informative at the microscopic level of study. This is due to a number of properties of the neutron that are essential for its interaction with a solid.

The first is the spin of the neutron, which ensures that it has a magnetic moment. Due to this, the neutron interacts with magnetic moments in matter with approximately the same efficiency as with the nuclei of atoms of matter due to nuclear forces. This makes it possible to study both the spatial distribution and the dynamics (i.e., spectral characteristics) of magnetic moments in a substance using the same instruments (diffractometers and spectrometers) that are traditionally used to study the atomic structure and dynamics of solids.

The interaction time parameters of neutrons with matter are quite unique in relation to the physical characterisation of the intermediate-valence state of rare-earth atoms.

The characteristic interaction time is $10^{-11}$–$10^{-13}$ s, which makes it possible to study in detail the evolution of spin fluctuations as function of external factors for heavy-fermionic and intermediate-valence compounds.

Thus, the use of magnetic scattering of thermal neutrons and, especially, neutron spectroscopy makes it possible to study the specific of excitation spectra and corresponding physics in f-electronic systems, as well as f–f interaction in metals. Namely, these properties underlie the physics of rare-earth systems with strong electronic correlations. The possibilities of neutron spectroscopy in relation to the physics of such systems are described in more detail in the reviews [30–32].

This review is organised as follows:

* Section 2 of the review examines the essence and theoretical foundations of the formation of long-range magnetic order in systems with a singlet ground state—the phenomenon of induced magnetism. The main experimental results are presented for an intermetallic compound PrNi in which the induced magnetically ordered state is realised. Furthermore, the magnetic phase diagram of PrNi under conditions of substitution in the rare-earth sublattice is examined;
* Section 3 convincingly demonstrates the simultaneous presence of a homogeneous intermediate valence state and long-range magnetic order in systems based on $EuCu_2(Si,Ge)_2$. The potential causes for the emergence of a magnetically ordered intermediate valence state in Eu are examined;
* In the conclusions (Section 4), the results presented in this review are analysed from the perspective of the "triangle of fundamental interactions".

## 2. Induced Magnetism in Singlet Ground State Systems

### 2.1. Model Consideration

The common magnetism observed in rare-earth intermetallics is attributed to the alignment of localised magnetic moments, which result from indirect RKKY exchange interactions. It should be noted that localised magnetic moments exist even in the paramagnetic phase, but their average spatial and temporal values are zero. The state of long-range magnetic order is formed as the temperature decreases, at which point the average value of the magnetic moment becomes non-zero. In this case, the temperature of magnetic ordering is determined by the magnitude of the exchange interaction parameter.

In fact, the formation of magnetically ordered states in a range of rare-earth sublattices presents a more intricate scenario. The effects of the crystalline electric field (CEF) play a crucial role in determining the nature of magnetic ordering for these compounds. At temperatures approximately equal to the energy splitting in the CEF (typically ranging from $10^{-3}$ to $10^{-2}$ eV), the crystalline field can reduce or even destroy the orbital contribution to the total ion moment, thereby suppressing the overall magnetic moment of the system. Thus, for rare-earth ions, the magnetic ordering arises from the competition between two factors: the effects of the crystalline electric field, which aim to suppress the ion moment and, consequently, destroy the magnetic order, and the exchange interaction, which strives for the opposite effect.

Against this backdrop, a nontrivial situation can arise, where long-range magnetic order emerges in a system where the crystalline field fully suppresses the magnetic moments through the formation of singlets in the ground state. This phenomenon has been theoretically predicted in several studies [9–11] and is referred to as "induced" magnetism. A characteristic feature of induced magnetism is the absence of magnetic moments due to the CEF effects, meaning that the ground state in the disordered phase is singlet, and local magnetic moments appear simultaneously with magnetic ordering.

Induced magnetism was initially analysed within the framework of a two-singlet model [9–13], which is visualised in Figure 1 and can be summarised as follows. Let us consider a scenario where there is an ion with two singlet levels (ground and excited) at any site of the crystal lattice. These levels are separated by the energy $\Delta$. The ground and excited states are connected by a non-zero dipole matrix element M. For rare-earth

ion systems, such a model appears physically justified: in compounds containing these elements, a situation can arise where the energy separating the ground and first excited crystal field split levels ($\Delta = E_1 - E_0$) is of the order $10^1$ K, while the other levels are much higher in energy (on the order of $10^2$ K). Consequently, at temperatures corresponding to magnetic ordering, the higher-lying CEF levels remain unoccupied and can be disregarded. Suppose these ions occupy equivalent sites in each elementary cell of the crystal. It is clear that, if there is no interaction between the ions, the system will exhibit paramagnetic behavior at all temperatures, even approaching absolute zero.

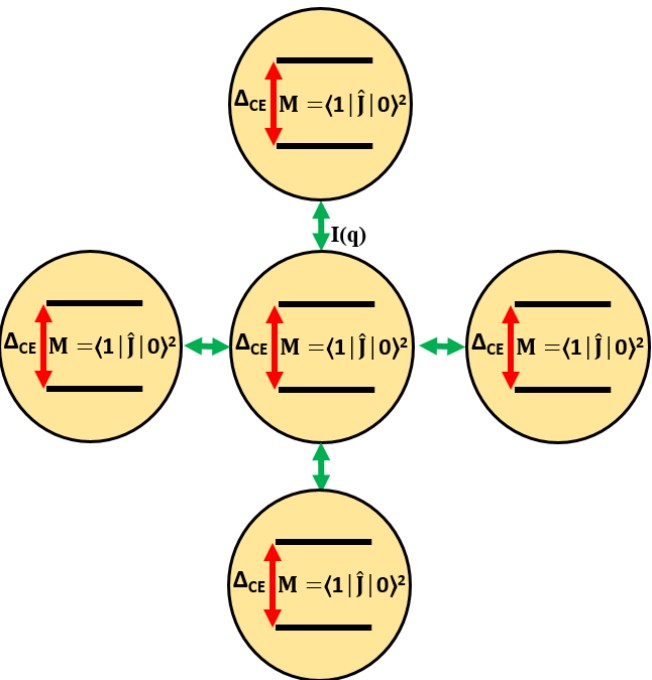

**Figure 1.** Visualisation of the two-singlet model of induced magnetism. Yellow circles correspond to rare-earth ions in the nodes of the crystal lattice. The exchange interaction I(q) of each ion with its neighbours is indicated by green arrows. For each ion, splitting into two singlets due to the CEF effects is observed: the ground state and the first excited state with a dipole matrix element M coupling these states.

The situation undergoes a fundamental change with the introduction of interionic interactions, for example, in a simple form of lattice sum over neighbors:

$$I(\mathbf{q}) = \sum_{ik} I_{ik} e^{-i\mathbf{q}\mathbf{r}_{ik}}; \tag{1}$$

here, $I_{ik}$ is the exchange integral between the rare-earth ions located at the lattice sites $i$ and $k$, $\mathbf{q}$ is a reduced wave vector, and $\mathbf{r}_{ik}$ is the radius vector. The dipole matrix element, in combination with the interionic interaction, influences the wave functions by competing with the crystalline field. Consequently, the collective action of these forces can generate a polarisation factor, leading to the emergence of induced magnetic moments at each lattice site and the simultaneous orientational ordering of these moments. It has been shown [11] that the fulfillment of a critical parameter condition is necessary for the formation of an ordered magnetic state in such a system:

$$A_2 = \frac{2M^2 I(\mathbf{q}_{cr})}{\Delta} \geq 1 \tag{2}$$

If the condition for the critical parameter is satisfied, then, at a certain point in the Brillouin zone, the energy of elementary excitations becomes zero $\omega(\mathbf{q}_{cr}) = 0$, resulting in

the emergence of induced ferro-($\mathbf{q}_{cr} = 0$) or antiferromagnetism ($\mathbf{q}_{cr} = \frac{\pi}{a}$). The temperature of magnetic ordering within the framework of the two singlet model is determined by the following expression:

$$\tanh \frac{\Delta}{2T} = \frac{\Delta}{2M^2 I(\mathbf{q}_{cr})} \tag{3}$$

Let us discuss the extensions of the two-singlet model for induced magnetism. The three-singlet model of induced magnetism for the case of two magnetic ions in a primitive cell was developed in the study [33]. Using a mean field in random phase approximation, the energy dispersion relations are obtained for the magnetic excitations above ordering temperatute along the high symmetry directions. For the case where there is no dipole interaction between two excited states $E_1$ and $E_2$ (i.e., the matrix element $M_{12}$ for the dipole transition $E_1 \rightarrow E_2$ is zero), the expression for the critical parameter, which determines the transition to an ordered state, is given by

$$A_3 = \frac{2M_{10}^2 I(\mathbf{q}_{cr})}{\Delta_{10}} + \frac{2M_{20}^2 I(\mathbf{q}_{cr})}{\Delta_{20}} \geq 1 \tag{4}$$

where $M_{\alpha 0}$ and $\Delta_{\alpha 0}$ are dipole magnetic matrix elements of transitions and the energy distance between singlet levels (ground state—0 and excited states $\alpha = 1, 2$), respectively.

Equation (4) can conveniently be used as a condition for the exchange interaction $I(\mathbf{q}_{cr})$—the transition to an ordered magnetic state is possible only when the exchange interaction exceeds a critical value $I(\mathbf{q}_{cr}) > I_{cr}$, which is determined by the energy of the singlet levels and the transition matrix elements:

$$I_{cr} = \frac{\Delta_{10}\Delta_{20}}{2(M_{01}^2 \Delta_{20} + M_{02}^2 \Delta_{10})} \tag{5}$$

In the theoretical study [34], a detailed analytical treatment of induced moment behavior in the physically important three-singlet model relevant for non-Kramers f-electron systems in lower than cubic symmetry was proposed. The study derived the condition for the emergence of an ordered magnetic state and investigated the exciton mode dispersions using the random phase approximation response function formalism and the Bogoliubov quasiparticle picture for such a system.

### 2.2. Experimental Results and Further Development of the Model

One of the early attempts to systematically investigate induced magnetism was conducted on the single crystal of the compound $Pr_3Tl$. This system features a singlet ground state and exhibits a ferromagnetic ordered state below an ordering temperature of $T_C = 11$ K. The dispersive nature of excitations in the crystalline electric field was discovered through a series of studies involving neutron scattering on $Pr_3Tl$ samples [35,36]. Contrary to the predictions of the induced magnetism model, a strong temperature dependence of the dispersion was not observed, and the transition to an ordered state through a soft magnetic mode was not confirmed. The presumed reason for this discrepancy [37,38] is the more complex scheme of the CEF levels. It involves a ground state singlet, a first excited state triplet, and other closely spaced energy-level states with non-zero transition matrix elements. This complexity arises due to the high local crystallographic symmetry (cubic) of the Pr ion positions in $Pr_3Tl$. Additional experiments investigating quasielastic magnetic neutron scattering [39] have linked the magnetic ordering to the formation of a central peak (E = 0) at temperatures approximately equal to the ordering temperature. Further attempts to detect experimentally induced magnetism have been realised on other rare-earth metal compounds characterised by a singlet ground state and undergoing magnetic ordering ($PrSb$ [40], $TbSb$ [41], $PrCuO_4$ [42], $Pr_5Ge_4$[43], etc.). However, these attempts have also been unsuccessful: in all of these systems, it was not possible to fully satisfy the conditions of the model given in the Equation (2).

The investigation of induced magnetism in the compound PrNi has proven to be considerably more successful. The intermetallic compound PrNi crystallises in the orthorhombic CrB-type structure (space group Cmcm) (see Figure 2a). There is one crystallographic type (4c) of site for Pr. This results in the presence of two Pr ions per primitive cell, forming two identical sublattices. The local symmetry of the Pr site (point group $C_{2v}$) completely removes the degeneracy of the ground 4f-electron multiplet (J = 4). The CEF interaction splits this multiplet into nine singlets.

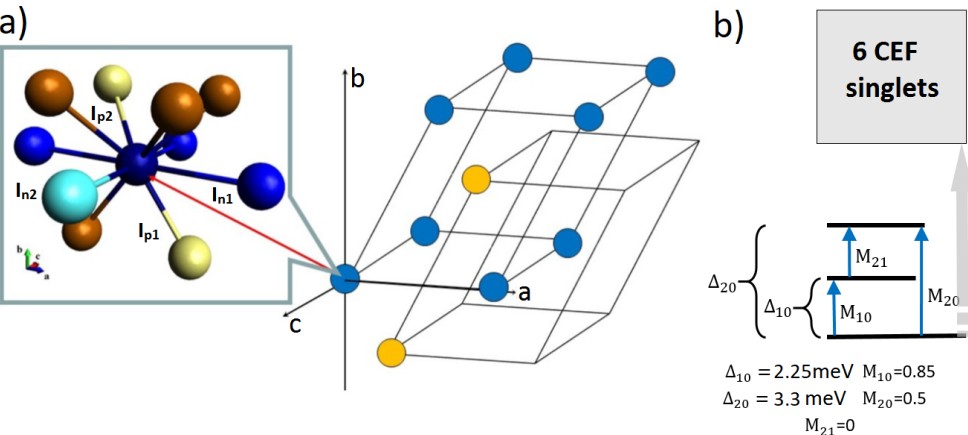

**Figure 2.** (**a**) (from [33]) The structure of the nearest environment among the RE occupied positions in the PrNi. There are two RE-ions in the primitive cell, and two equivalent RE-sublattice are shown in blue and bronze. Insert: the structure (see Section 2.3) of the nearest neighbors configuration with 10 RE ions in two coordination spheres involving both sublattices. Additionally shown is 1 variant of 18 possible occupations for RE-ion positions by Pr and La for the particular composition $(Pr_{0.7}La_{0.3})Ni$ (blue color corresponds to sublattice n, bronze color corresponds to sublattice p, according to Equation (1), $I_{ij}$ are exchange constants). La ions are marked with light color. (**b**) Splitting scheme in the crystal electric field for PrNi [13,33]. The allowed magnetic dipole transitions are indicated by arrows.

Thus, the ground state in PrNi is necessary for a singlet due to the low symmetry at the Pr site. Nevertheless, experiments measuring magnetic susceptibility [44] and neutron diffraction [13] have revealed that this compound undergoes ferromagnetic ordering at $T_C$ = 21 K (see inset in Figure 3a). The neutron inelastic scattering experiments were conducted on a single crystal sample of PrNi [13]. The typical spectrum for the **Q**∥[010] direction is shown in Figure 3a. It was found that the spectrum exhibits two modes of magnetic excitations, whose energy positions are significantly dependent on **q**. As the temperature was lowered to T = 21 K, there was a gradual reduction in the lower energy excitation frequency, nearly approaching zero at q = 0 (the center of the Brillouin zone), as depicted in Figure 3a.

The total intensity of two magnetic excitations from the ground state to $E_1$ and $E_2$ observed in the neutron spectrum [45] is 7.5 ± 0.5 barn, i.e., close to the total magnetic cross-section for $Pr^{3+}$. This means that coupling of the ground and two excited states with the remaining six levels is extremely weak, and up to 95% of the total neutron cross-section for magnetic scattering is related to two transitions $M_{10}$ and $M_{20}$ from ground (0) to excited (1, 2) states. In fact, the real situation for nine-fold splitting of $Pr^{3+}$ state in PrNi is reduced to three singlets (the ground one and two excited ones), where these excited states both are dipole-linked only to the ground state, as shown in Figure 2b.

Theoretical investigations into induced magnetism in PrNi were conducted using the three-singlet model [33]. The values of the matrix elements $M_{10}$ and $M_{20}$ were obtained from the analysis of the spectra of inelastic magnetic neutron scattering for PrNi ($M_{10}$ = 2.9, $M_{20}$ = 2.1) [46]. Only two coordination spheres of the nearest neighboring Pr ions in each sublattice (exchange interaction constants $I_{n1}$ and $I_{n2}$) and between the sublattices

(exchange interaction constants $I_{p1}$ and $I_{p2}$) were considered for the calculation of Pr–Pr interaction. The values of exchange interaction constants were obtained by fitting the dispersion along the **Q** ∥ [100] direction.

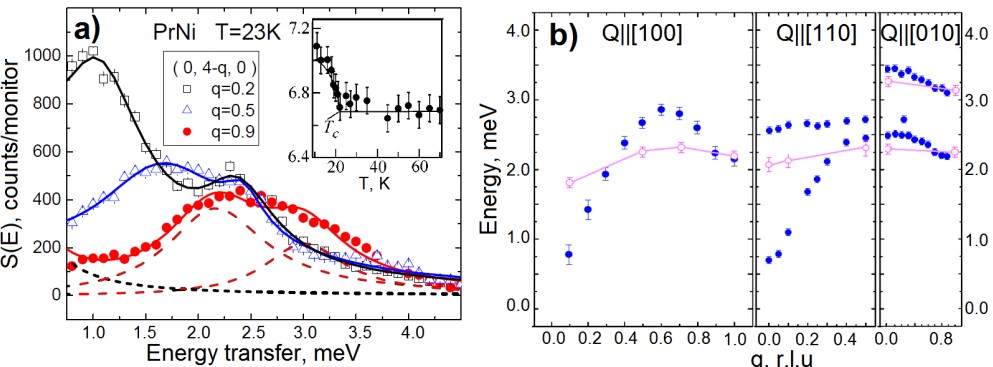

**Figure 3. (a)** (from [45]) Experimental neutron spectra of a PrNi single crystal at T = 23 K were measured for **Q**∥[010]. For the spectrum with q = 0.9, the contribution from magnetic scattering is indicated by a dashed line, while the background is represented by a dotted line. The inset shows the temperature dependence of the intensity of the (200) Bragg peak in the neutron diffraction pattern. **(b)** (from [13]) The dispersion relation of magnetic excitations in PrNi is depicted. The filled symbols correspond to measurements at T = 23 K, while the open symbols connected by a line represent data at T = 70 K.

Figure 4 presents a comparison between the calculated dispersion curves and the experimental results obtained from [13]. The experimental data include all observed branches (with non-zero intensities) measured at a selected reciprocal lattice site. The three-level model demonstrates improved agreement compared to the two-level model employed in [13], without the need for additional exchange constants. Both the qualitative and quantitative agreement between the calculated curves and the experimental data [13,45] across all Q directions were enhanced, despite the fact that the fitting parameters were determined using only one direction, **Q** ∥[100]. The decrease in energy and a significant increase in the intensity of the lower (acoustic) mode near q = 0 confirmed the validity of the theoretical concept of "induced magnetic order". This observation provides an interpretation for the magnetic phase transition observed in PrNi, which is driven by the influence of the "soft mode".

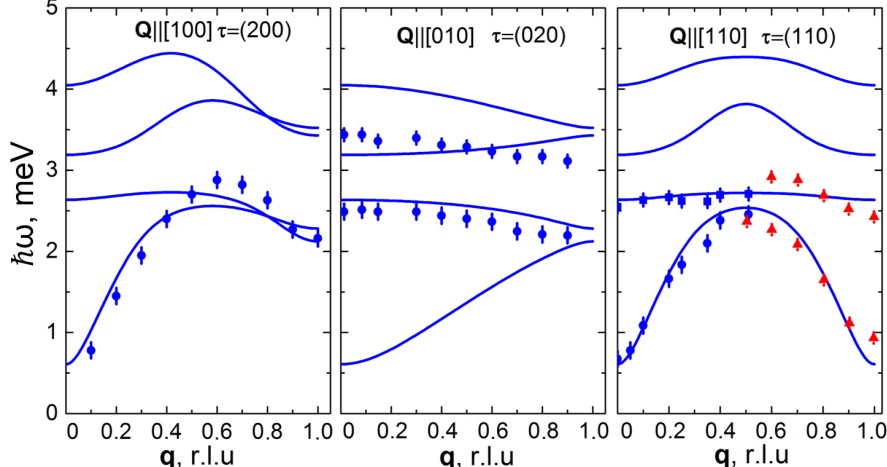

**Figure 4.** Comparison of calculated (lines), according to the three-singlet model of induced magnetism for the case of two magnetic ions in a primitive cell [33], and experimental (blue points [13] and red triangles [45]) energy dispersions of magnetic excitations in PrNi at T = 23K.

### 2.3. Phase Diagram for PrNi with Substitution

A detailed study of the characteristics and driving forces of such an exotic form of magnetism can be realised by investigating the dependence of the magnetic ordering temperature on variation of competing interactions. These variations can be achieved, in particular, by introduction of defects into the magnetic sublattice of compounds. The defects can affect the magnetic ordering temperature in different ways, which is clearly related to the nature of both the magnetism itself and the cooperative state in particular systems.

The simplest defect that allows for the investigation of the role of exchange interaction in the formation of the magnetic state of a system is a rare-earth ion with zero magnetic moment substituting the basic rare-earth ion with magnetic moment at its lattice site, labelled as a "magnetic hole" or "non-magnetic impurity". Several studies on the influence of non-magnetic impurities on the ordering temperature in systems with magnetism driven by direct exchange interaction have shown a linear relationship between the ordering temperature $T_C$ and the average magnetic moment at the lattice site [47–49]. A similar correlation between the magnetic ordering temperature and the parameter of indirect exchange interaction has been observed in systems with RKKY magnetism. A clear example can be seen in the results of studies [50,51] on the ferromagnet with RKKY-type exchange interaction in GdNi. In this compound, Gd ions with purely spin magnetic moment (L = 0) were substituted with non-magnetic ions La, resulting in a decrease in the effective exchange interaction parameter. It has been established that, with such substitution, the magnetic ordering temperature linearly decreases with decreasing concentration of magnetic ions and, consequently, weakening of the exchange interaction. The transition to the ordered magnetic state for $Gd_{1-x}La_xNi$ is observed at concentrations of non-magnetic ions La $x < 0.9$.

In the case of the PrNi system, the substitution of Pr ions with La ions is considered a "magnetic hole" type of substitution. It has been observed [45] that the substitution of Pr with La has a weak impact on the lattice parameters and, therefore, does not significantly alter the crystal field potential parameters. Thus, the substitution of Pr ions with La ions in PrNi primarily affects the exchange interaction while leaving other parameters of the system largely unchanged.

The dependence of the magnetic ordering temperature on the concentration of La impurity in $Pr_{1-x}La_xNi$ was determined in the study [33] through measurements of AC susceptibility and DC magnetisation. The measurement results for concentrations $x = 0, 0.2$, and $0.5$ are presented in Figure 5 by black squares. It has been found that the introduction of La ions with zero magnetic moment (effectively introducing a "magnetic hole") into the induced magnetism system PrNi leads to a linear decrease in the ordering temperature down to T = 0 at a La concentration of $x \approx 0.5$. This is significantly different from systems with RKKY-type magnetic ordering, where magnetic ordering was observed for up to 90% non-magnetic impurities.

The influence of non-magnetic impurities on the magnetic ordering temperature in PrNi has been explained within the framework of the Microscopic States Model (MSM) [33]. The essence of the MSM lies in considering the actual nature and structural characteristics of the distribution of magnetic interactions in the rare-earth sublattice when impurity ions are introduced. Non-magnetic substitution in PrNi results in the following. The value of the exchange constant $I_{ij}$ is equal to 0 for all atoms ($La^{3+}$) randomly distributed in the $Pr_{1-x}La_xNi$ crystal lattice. This makes it possible to simulate all possible configurations and, consequently, the exchange parameters for the cluster of 10 rare-earth atoms with random Pr/La substitution. The inset in Figure 2 schematically presents one of the possible configurations for the substitution parameter $x = 0.3$, illustrating the nearest environment of the Pr ion within two coordination spheres. It is important to note that these configurations contribute to the formation of magnetic order only when the exchange parameter exceeds the critical value $I(\mathbf{q}_{cr}) > I_{cr}$ (5). The calculation results obtained within the MSM are in good agreement with the experimental data, showing a clear linear dependence of the ordering temperature on the concentration of non-magnetic substitution (dashed line in Figure 5).

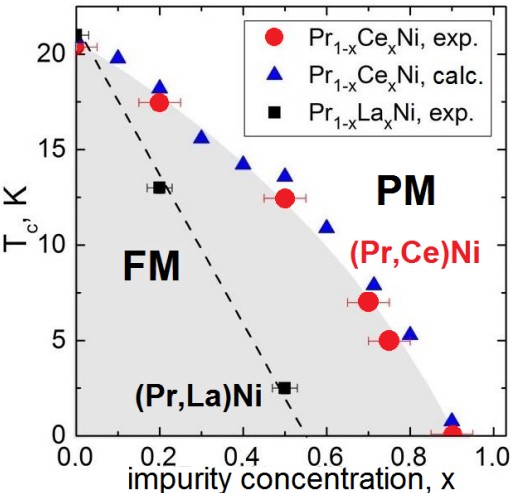

**Figure 5.** The relationship between $T_C$ and $x$ in $Pr_{1-x}La_xNi$ (black squares) [33] and $Pr_{1-x}Ce_xNi$ (red circles) [52]. The blue triangles represent the $T_C$ calculations for $Pr_{1-x}Ce_xNi$ using a microscopic state model. The calculation result for $Pr_{1-x}La_xNi$ is presented as dashed line. The grey area indicates the region of the ferromagnetic state of the (Pr,Ce)Ni system, which is significantly larger than for (Pr,La)Ni.

The influence of magnetic ion substitutions with intermediate valence ions on the induced LRMO was investigated in the study [52]. For the intermetallic system GdNi with an RKKY-type magnetism, the impact of Ce impurities on the magnetic ordering temperature was examined in studies [50,51]. Surprisingly, the experiments revealed an intriguing fact: the substitution of Gd ions with non-magnetic La and intermediate valens state Ce impurities had an essentially equal effect on the magnetic ordering temperature throughout the entire range of substitution concentrations. The values of $T_C$ at equal substitution concentrations were within the range of experimental error. The presence of spin fluctuations whose energy exceeds $10^2$ K is characteristic of the intermediate valence phenomenon [32,53]. It was expected that the intermediate valence would have a significant influence on the induced magnetism through the transformation of exchange interactions due to strong spin fluctuations on the Ce ions, which are introduced into the lattice of induced local magnetic moments.

The influence of Ce impurities on $T_C$ in PrNi, as investigated in [52], significantly differs both from the effect of La impurities (Figure 5) and from the influence of impurities on the RKKY magnetism in GdNi. For $Pr_{1-x}Ce_xNi$, the concentration dependence is much weaker, as indicated by the circles in Figure 5, and the possibility of ordering ($T_C > 0$) is preserved up to $x_{Ce} = 0.85$. Calculations within the MSM for $Pr_{1-x}Ce_xNi$ have revealed that the main influence on the ordering temperature due to the substitution of Pr with the intermediate-valence ion Ce is attributed to a significant suppression of crystal field splitting for $Pr^{3+}$, as observed in [52]. According to experimental observations [52], it has been noticed that the energy of the first excited state in the $Pr_{0.25}Ce_{0.85}Ni$ sample decreases compared to the PrNi system, going from 2.2 meV to 0.8 meV. On the other hand, the energy of the second excited level increases from 3.2 meV to 4.6 meV. Based on the findings of the study [54], it has been suggested that the renormalisation of the CEF splitting is linearly related to the concentration of IV ions of Ce. This means that the magnetic moment of the Ce ion does not manifest itself due to its suppression caused by fast spin fluctuations, which is consistent with experimental results for Gd(La,Ce)Ni systems. Only taking int account CEF effects results in the correspondence between MSM calculations and experiments. The calculation for the whole range of Ce concentrations in $Pr_{1-x}Ce_xNi$, taking into account the renormalisation on CEF splitting for $Pr^{3+}$, was fulfilled in the framework of MSM [52]; the calculation results were in quantitative agreement with the experimental data (see Figure 5).

The influence of competing exchange interactions and crystal field splitting on the induced magnetic ordering states in PrNi-based systems is visually presented in Figure 6.

The ground state diagram was obtained through calculations within a model of three singlets [52] with fixed values of the dipole matrix elements, corresponding to the case of PrNi and $\Delta_{20} \rightarrow \infty$. Point 1 corresponds to the parameter values I($q_{cr}$) and $\Delta$ characteristic of PrNi. Upon substituting Pr with non-magnetic La ions, according to the MSM, the value of the parameter I($q_{cr}$) decreases while $\Delta$ remains constant. The change in the ground state type during the Pr to La substitution from ferromagnetic ($T_C$ = 21 K) to paramagnetic is indicated by the solid arrow (1–2–3). The possibility of transitioning to an ordered magnetic state is preserved up to La concentrations of $x$ = 0.5, which is in good agreement with experimental observations [33]. In $Pr_{1-x}Ce_xNi$, both I($q_{cr}$) and $\Delta$ decrease with increasing Ce concentration (dashed arrow in Figure 6). The decrease in the exchange parameter is "compensated", according to Equation (4), by the decrease in the energy gap between the CEF levels. As a result, the ordering temperature decreases much more slowly with increasing $x$ (e.g., point 4 corresponds to $x$ = 0.8) compared to the La substitution case.

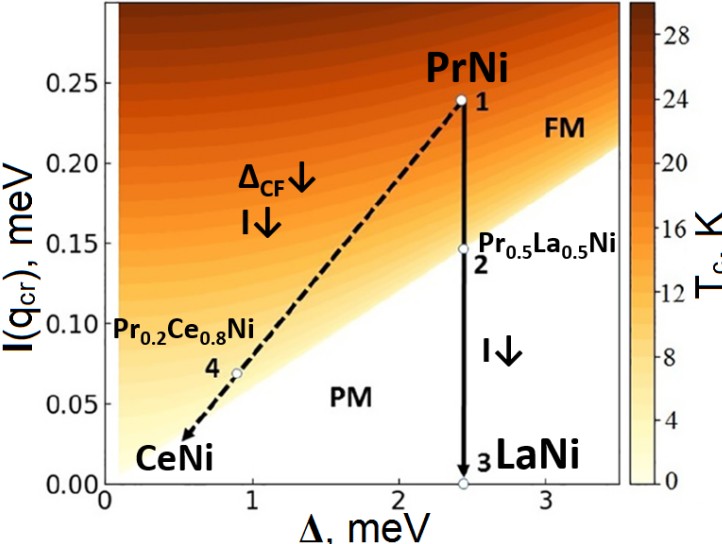

**Figure 6.** (from [52]) The ground state diagram for systems based on PrNi is depicted, illustrating the relationship between the magnetic ordering temperature ($T_C$), the exchange parameter I($q_{cr}$), and the splitting between CEF singlets $\Delta$. The solid line represents the variation in $T_C$ with increasing La concentration in $Pr_{1-x}La_xNi$, while the dashed line represents the change in Tc with increasing Ce concentration in $Pr_{1-x}Ce_xNi$. The labeled points correspond to the following compounds: 1. PrNi, 2. $Pr_{0.5}La_{0.5}Ni$, 3. LaNi, 4. $Pr_{0.2}Ce_{0.8}Ni$.

## 3. Homogeneous Intermediate Valence and Long-Range Magnetic Order

### 3.1. TmSe—The First Example of the Coexistence of LRMO and IV

The first system in which the combination of intermediate valence for rare-earth ion and long-range magnetic order was experimentally demonstrated is TmSe. The valence value of Tm in this compound is weakly temperature dependent and amounts to $v_{Tm} \approx 2.6$ [15,16]. In TmSe, a long-range magnetic order with a relatively small magnetic moment ($m_{eff}$ = 1.7 $\pm$ 0.2$\mu_B$, which is significantly lower than in the case of $Tm^{2+}$ ($m_{eff}$ = 4.5$\mu_B$) or $Tm^{3+}$ ($m_{eff}$ = 7.5$\mu_B$)) is established at temperatures below $T_N$ = 3.45 K. The effects of the crystalline electric field in this compound were disregarded because of the energy scale of the spin fluctuations [55].

Extensive neutron scattering studies have been perfomed on both polycrystalline [16] and single-crystal [56,57] samples of this compound. The main results of the experimental measurements are presented in Figure 7. At high temperatures, the magnetic excitation spectrum of TmSe exhibits typical behavior observed in intermediate-valence compounds: a nearly temperature-independent broad quasielastic signal corresponding to spin fluctuations (Figure 7) with a linewidth of $\Gamma_{QE}/2$ = 6 meV is observed. However, the temperature evolution of the magnetic excitation spectrum in TmSe is atypical for intermediate-valence

compounds. Reducing the temperature to 100 K leads to a significant change in the spectrum: an inelastic component appears around E = 10 meV, and its intensity increases as the temperature decreases further. The width of the quasielastic peak decreases without exceeding the value of $\Gamma_{QE} = k_B T$. Below the magnetic order temperature $T < T_N$, the quasielastic signal is not detected and, at an energy of 1 meV, an inelastic peak associated with magnon excitations appears [16].

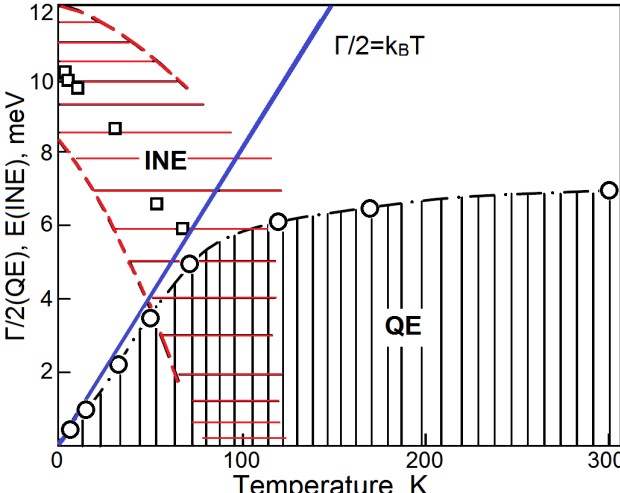

**Figure 7.** (from [16]) The temperature dependence of the quasielastic linewidth (circles) and the energy of the inelastic line (squares) are shown [16]. The width of the inelastic line in the TmSe neutron spectrum is indicated by a hatched band, bounded by dashed curves.

The inelastic component of the neutron scattering spectrum in TmSe was thoroughly investigated in [57]. It has been established that the inelastic component of the spectrum exhibits significant energy dispersion, and its intensity is modulated in the Brillouin zone. The maximum intensity and minimum energy are observed near the Brillouin zone boundary. The influence of the substitution of Tm by non-magnetic ions Y and La on the inelastic neutron scattering spectra of TmSe was investigated in [58]. It was found that the inelastic component of the spectrum around E = 10 meV was observed at low temperatures, even in highly diluted samples (with a concentration of substitution atoms up to 95% in $Tm_{0.05}Y_{0.95}Se$ and $Tm_{0.05}La_{0.95}Se$). This finding allowed the determination of the origin of the corresponding excitation as a single site.

Theoretical concepts regarding the formation of ordered magnetic states in the intermediate-valence system, consistent with the experimental results presented above, were proposed in the study [55]. The existence of an inelastic component in the magnetic excitation spectrum was explained by the presence of "time coherence" occurring on a single ion between two competing electronic Tm configurations, each carrying a magnetic moment ($4f^{12}$, J = 6 and $4f^{13}$, J = 7/2). Coherent fluctuations of this type have been supposed to not violate the RKKY interaction, allowing the system to be ordered at low temperatures. The energy dispersion and intensity of the inelastic component in the TmSe spectrum have also been described within the framework of the lattice model of time coherence [59,60]. It is important to note that the presence of magnetic moments in each of the competing configurations is a key element of the theories presented.

### 3.2. Eu-Compounds—Main Features

The study of $EuCu_2(Si,Ge)_2$-type systems seems to be very important in terms of the coexistence of intermediate valence and LRMO. For this series of compounds [17–22], as well as for similar 1-2-2-type Eu systems [23,25–29], extensive regions of the phase diagram have been obtained, corresponding to both the ground state with LRMO AFM-type and the intermediate-valence state of europium. An important feature of all these compounds is that one of the initial Eu configurations involved in the formation of the

intermediate-valence state is formally non-magnetic ($Eu^{3+}$, J = 0), similar to Ce- and Yb-based systems, which do not exhibit LRMO [14,32]. Nevertheless, for a number of $EuCu_2(Si,Ge)_2$ systems in a wide range of compositions (Ge concentrations), an ordered magnetic state was detected by thermodynamic measurements [19] and then directly confirmed by neutron diffraction [17].

It is important to note that in the "classical" intermediate-valence compound $SmB_6$ ($\nu \approx 2.5$), one of the "competing" configurations ($Sm^{2+}$: $^7F_0$, J = 0), is nonmagnetic, similar to $Eu^{3+}$, and the mechanisms of IV state formation in $SmB_6$ and $EuCu_2Si_2$ are fundamentally similar [61]. However, in related series based on $SmB_6$ [14,62] with different Sm valencies, coexistence of LRMO and the intermediate-valence states has not been observed. It is important to note that, in contrast to the Eu-based 1-2-2 systems, recent studies have shown that the valence variation with pressure is quite weak for $SmB_6$ [63].

Let us focus on discussing $EuCu_2(Si_xGe_{1-x})_2$, as it is one of the most extensively studied Eu-based compounds. During the initial phase of the study of $EuCu_2(Si_xGe_{1-x})_2$, it was discovered [64] that, when silicon is substituted by germanium at certain silicon concentrations ($0.6 < x < 0.8$), there is an observed temperature dependence of the resistivity and thermoelectric power characteristic of Kondo compounds. Detailed investigations of the thermodynamic, magnetic, and kinetic properties of solid europium-based substitutional alloys, $EuCu_2(Si_xGe_{1-x})_2$, were carried out in the study [19]. Evidence for a magnetically ordered ground state has been found at silicon concentrations $x < 0.65$. Further increases in silicon concentration lead to a change in the nature of the ground state, with a heavy fermion ground state forming in the range $0.7 < x < 0.8$. X-ray spectroscopy experiments conducted in [20] have allowed the determination that the valence of europium is non-integer for all silicon concentration values.

The compounds $EuCu_2(Si_xGe_{1-x})_2$ have been studied using neutron experimental techniques, confirming the results obtained from thermodynamic studies. The appearance of additional magnetic peaks in the neutron diffraction pattern for several silicon concentrations ($x = 0, 0.4, 0.6$) reliably established the presence of an antiferromagnetic ordered state in these compounds [65]. In addition, the tendency is observed for the europium ordered magnetic moment to decrease with increasing silicon concentration (at a temperature of T = 5 K, $M_{x=0} = 6.7\mu_b$, $M_{x=0.4} = 5.3\mu_b$, $M_{x=0.6} = 5.3\mu_b$). Inelastic neutron scattering experiments have confirmed the presence of a heavy fermion regime in the ground state of $EuCu_2(Si_xGe_{1-x})_2$ for $0.7 < x < 0.8$. It was found [17] that further increasing the silicon concentration up to complete germanium substitution leads to the formation of a low-temperature spin-gap state, which transfers to a state with strong spin fluctuations above 100 K (see Figure 8).

It is worth noting [17] that, at silicon concentrations $x > 0.4$, a relatively broad quasielastic signal is observed in the magnetic excitation spectrum at T $\geq$ 100 K, characterising the energy of spin fluctuations in the system. The quasielastic signal persists even at low temperatures, with a width well above $k_B T$. Therefore, the neutron experiments have reliably established [17] that the $EuCu_2(Si_xGe_{1-x})_2$ system exhibits a coexistence of an ordered magnetic state and strong spin fluctuations over a wide range of silicon concentrations. The final phase diagram, based on the combined results of neutron, X-ray, and thermodynamic experiments, is shown in Figure 8b.

Based on the above studies, the nature of the intermediate valence of Eu remains uncertain. The widely used method of substituting specific components in alloys, often referred to as "chemical pressure", is considered ambiguous due to the possibility of forming local inhomogeneities which could result in an inhomogeneous intermediate-valence state. The "purest" method of controlling the valence states of rare-earth ions in this sense is by applying hydrostatic pressure. This method does not introduce any structural or electronic defects into the crystal lattice. In this situation, it was extremely important to make a direct comparison of the effects of hydrostatic and "chemical" pressure on the initial compound $EuCu_2Ge_2$. The phase diagram of the intermediate-valence system

$EuCu_2Ge_2$ has been studied by measuring the temperature dependence of the heat capacity and electrical resistance as a function of pressure up to 15 GPa in [22].

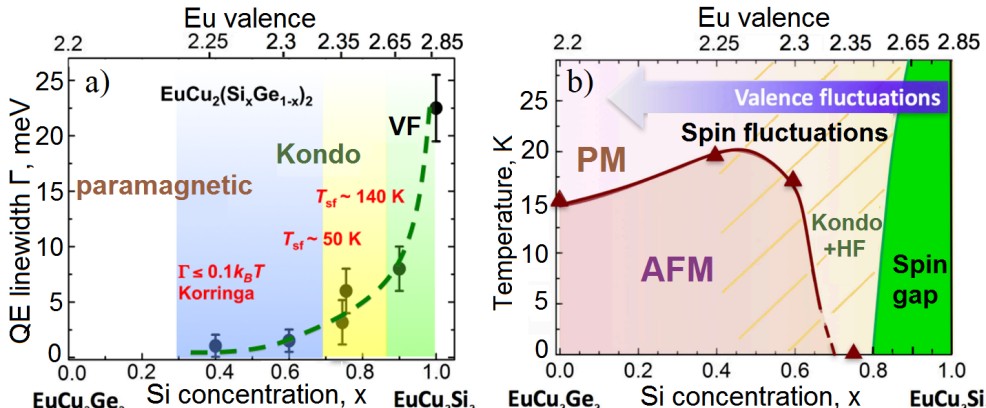

**Figure 8.** (from [17]) Some neutron scattering study results for $EuCu_2(Si_xGe_{1-x})_2$. (**a**) Concentration dependence of the magnetic quasielastic linewidth (FWHM) at T = 100–200 K for $EuCu_2(Si_xGe_{1-x})_2$ series (see text) [17]; (**b**) Magnetic phase diagram for $EuCu_2(Si_xGe_{1-x})_2$ based on the original data from Refs [17,19]. The valence values shown on the upper scale are derived from X-ray absorption near-edge structure (XANES) data at T = 7 K, obtained in refs. [17,18]. The solid brown line represents the phase boundary between the antiferromagnetic (AFM) and paramagnetic (PM) states. The triangles indicate the values obtained from neutron powder diffraction experiments. The yellow region (Kondo + HF), extending down to $x = 0.4$ and overlapping with the AFM region, represents the spin-fluctuation regime where a quasielastic response is observed. The green colored region represents the low temperature spin gap regime.

The dependence of $T_N$ on pressure according to the results of all the above experiments is shown in Figure 9. The dependence of $T_N$ on pressure obtained in [22] qualitatively and quantitatively (by the value of $T_N$) reproduces the dependence of $T_N$ on the concentration of Si shown in the inset in Figure 9. The monotonic increase in the ordering temperature with pressure turns into a sharp decrease; there is no magnetic ordering at P > 9.5 GPa, similar to Si concentrations above $x = 0.65$. Note that these results are in good agreement with recent studies on the effect of pressure on $T_N$ in $EuCu_2(Si_xGe_{1-x})_2$ systems ($x = 0.5$ [66,67], $x = 0.45, 0.6$ [21]). Their observations show that an increase in Si concentration leads to a decrease in the critical pressure required to transfer the system to a paramagnetic state. The consistency of the dependence of $T_N$ on both $x$ and P confirms that a homogeneous intermediate-valence regime is achieved in the $EuCu_2(Si,Ge)_2$ system, similar to $EuCu_2Ge_2$ under pressure. This confirms the coexistence of intermediate valence and LRMO for the rare-earth sublattice as well. The valence appears as a universal parameter characterising the microscopic state of the system, which makes it possible to relate the effects of pressure and Si substitution on the Eu 1-2-2 system physical properties.

The direct relationship, established in [22], between the effects of "chemical" [18,20] and hydrostatic [68] pressure allows the generalisation of the results obtained from a series of neutron experiments and their correspondence to the magnetic phase diagram of $EuCu_2(Si,Ge)_2$. Lots of important data obtained from measurements of neutron spectra in relation to the magnetic phase diagram of $EuCu_2(Si,Ge)_2$ [17–19] are shown in Figure 10.

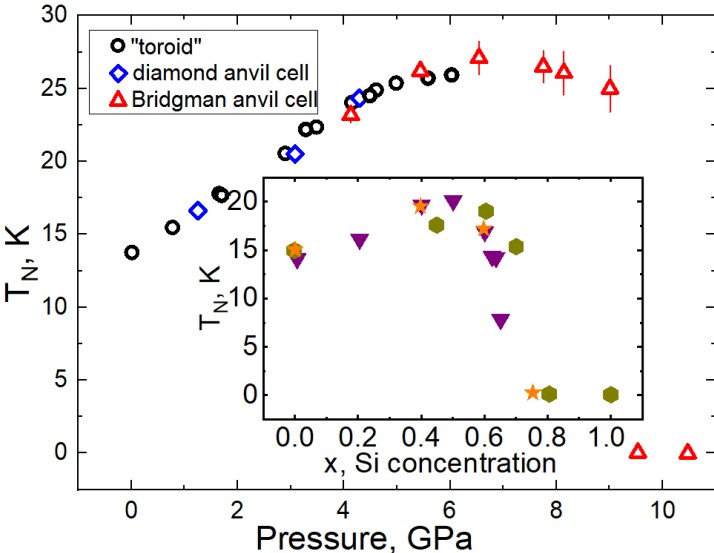

**Figure 9.** (from [22]) Pressure dependence of $T_N$ for $EuCu_2Ge_2$ based on the results of the series of experiments (experiment titles correspond to the designations adopted in ref. [22]). Inset: magnetic phase diagram of $EuCu_2(Si_xGe_{1-x})_2$ based on the heat capacity and resistivity measurements (downward triangles) [19] and (hexagons) [21] and on magnetic neutron diffraction data from [17] (stars).

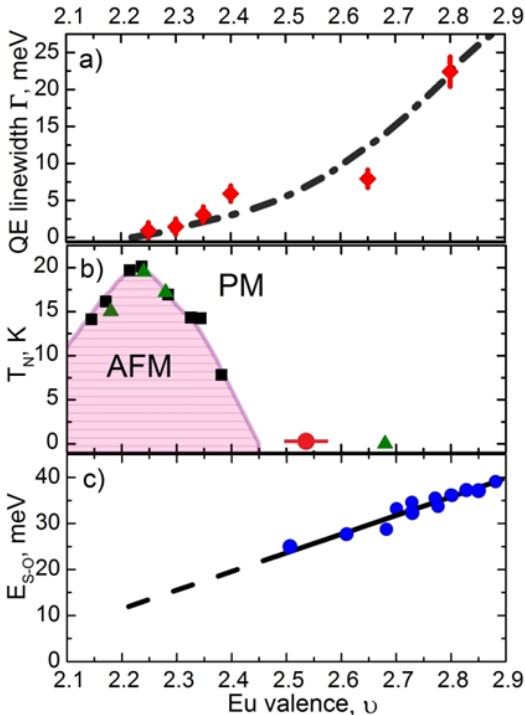

**Figure 10.** The parameters of the f-electron spectra obtained by neutron spectroscopy are analysed in relation to the magnetic phase diagram of the $EuCu_2(Si,Ge)_2$ compound. (**a**) Dependence of the quasielastic linewidth ($\Gamma_{qe}$) in the inelastic neutron magnetic scattering spectrum on the valence of Eu for the $EuCu_2(Si,Ge)_2$ series at T = 100–200 K [17]. The line is a guide to the eye. (**b**) The magnetic phase diagram for $EuCu_2(Si,Ge)_2$ is shown, incorporating the results of the study [22] (red circle) as well as the original data from ref. [19] (green triangles) and ref. [65] (black squares). The line represents the boundary between the antiferromagnetic (AFM) and paramagnetic (PM) phases. (**c**) The dependence of the spin-orbit transition energy ($^7F_0 \rightarrow {}^7F_1$) for the $EuCu_2(Si,Ge)_2$ series on the valence of Eu is determined by neutron spectroscopy [18]. The line is a guide to the eye.

Figure 10a shows the relationship between the valence and the width of the quasielastic magnetic neutron scattering peak ($\Gamma_{qe}$), which corresponds to the energy of the spin fluctuations. When the valence exceeds $\nu = 2.55$, a broad quasielastic signal with a width of about 2 meV ($\Gamma_{qe}/2 \sim 2$ meV) appears in the neutron spectra [17]. Remarkably, this signal persists even at very low temperatures (T $\sim$ 5 K) with a width (i.e., energy of the spin fluctuations) well above $k_b$T ($\sim$0.5 meV). At this point, the long-range magnetic order is already suppressed, as seen in Figure 10b, indicating the prominent influence of the hybridisation interaction on the electronic subsystem, starting from this particular value of the europium valence.

Figure 10c shows another important feature of neutron spectra—the valence-induced renormalisation of the spin-orbit excitation $^7F_0 \rightarrow ^7F_1$ (for free $Eu^{3+}$ with $f^6$ configuration, its energy is equal to 45 meV). Due to the spin-orbit splitting, the $Eu^{3+}$ ion (ground state $^7F_0$) has no magnetic moment at low temperatures and is a Van Vleck paramagnet (the states $J_0$ and $J_1$ are connected by a sufficiently strong magnetic dipole matrix element). The experimental dependence of the energy of this excitation on the Eu valence for a number of $EuCu_2(Si,Ge)_2$ systems, defined from magnetic neutron spectroscopy measurements, is shown in Figure 10c. It has been found [18] that, over the whole valence interval, the experimentally measured energy of the $^7F_0 \rightarrow ^7F_1$ spin-orbit transition decreases linearly with decreasing Eu valence: $E_{S-O}(\nu_{Eu} = 2.85) = 36$ meV, $E_{S-O}(\nu_{Eu} = 2.5) = 25$ meV.

It can be seen that the extrapolation of this dependence (dashed line in Figure 10c) leads to values close to zero of the transition energy in the region $\nu < 2.5$, which formally implies the formation of a degenerate ground state of the $f^6$ electron shell. This is a rather non-trivial phenomenon, but it follows from the clear systematic experimental observation. In fact, this allows us to conclude that the IV state of the RE subsystem in this region ($\nu < 2.5$) transforms into the IV state with the competition of two magnetic configurations instead of one magnetic ($Eu^{2+}$) and one nonmagnetic ("standart" $Eu^{3+}$).

It should be noted that, in a number of similar neutron experiments with other intermediate-valence compounds based on $SmB_6$ [57,61,69], where $f^6$ ($^7F_0$) configuration is involved in the formation of the IV state of Sm, no renormalisation of the spin-orbit splitting of the $f^6$ electron shell with a change in the valence of the rare-earth ion was observed. This suggests a fundamental difference [61] in the conditions for the formation of IV states in europium (due to hybridisation) and samarium (due to Coulomb interaction) compounds.

A generalised magnetic phase diagram as a function of the valence state of europium is shown in Figure 10b. By comparing it with the characteristics of the neutron spectra shown in Figure 10a,c, we can draw conclusions about the possible origin of the antiferromagnetic ground state region. This region is defined by two main factors: (i) the suppression of strong spin fluctuations and (ii) the presence of two magnetic configurations, namely, $f^7$ and $f^6$. For the $f^6$ configuration, the existence of a magnetic moment is assumed, in contrast to the Van Vleck paramagnetism exhibited by the free $Eu^{3+}$ ion. This is due to an unusual and as yet unexplained renormalisation of the spin-orbit splitting due to intermediate-valence phenomena.

It is possible that the transition to the ordered state occurs as the Eu valence decreases from values above 2.5, and this transition may be triggered by the formation of an induced magnetic moment in the initially non-magnetic $f^6$ configuration. This formation is influenced by the polarisation instability of the singlet ground state $J_0$ due to induced magnetic moment appearing in the singlet-triplet system ($^7F_0 \rightarrow ^7F_1$) for $f^6$ configuration in analogy with the CEF states of $Pr_3Tl$ compound [35,36]. Estimations based on the parameters of the neutron spectrum indicate that the critical parameter A (2), which relates to the decreasing energy of the spin-orbit splitting, can reach the value required for ordering in the considered Eu systems [70]. This suggests that the formation of an antiferromagnetic state in the system near $\nu \sim 2.5$ can be facilitated.

The presence of two magnetic configurations in a certain valence range corresponding to a wide range of concentrations $x$ makes the $EuCu_2(Si_xGe_{1-x})_2$ system similar to the TmSe. It has been suggested [55,58] that the phenomenon known as "time coherence", resulting from the competition between the two electronic configurations possessing magnetic

moments, could lead to the formation of an ordered magnetic state in the intermediate-valence system. It can be assumed that a similar mechanism of magnetic ordering exists in Eu systems, similar to the case of TmSe. However, further investigations [70] are needed to fully understand this aspect.

## 4. Conclusions

The formation of the uncommon magnetic order for systems with a singlet ground state is studied and analysed by the example of the rare-earth intermetallic compound PrNi. It has been shown that the paramagnetic–ferromagnetic phase transition occurs by the formation of a soft magnetic mode and is conditioned by the presence of the critical parameter. This parameter relates to three types of interactions: CEF splitting, magnetodipole coupling between corresponding f-electron states, and exchange interaction between rare-earth ions in a crystal. The influence of defects of different types ("magnetic hole", spin-fluctuating IV state) on the magnetic phase diagram is due to their influence on these competing factors. This can be calculated on the basis of developed model representations.

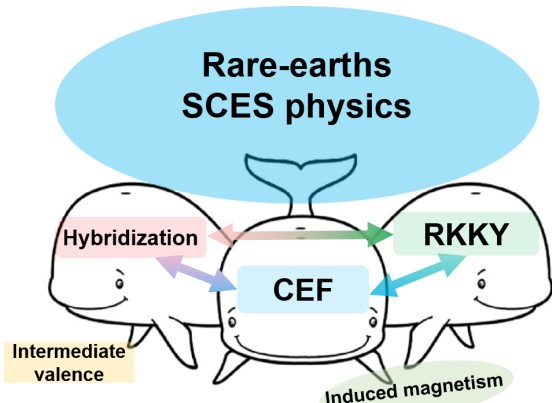

**Figure 11.** Diagram of the fundamental interactions characteristic of localised RE magnetic moments in solids. The arrows indicate their possible combinations. The CEF effects are inherent in all compounds of rare-earth elements; RKKY—exchange interaction between rare-earth ions, as well as hybridisation between localised f- and band electrons, are characteristic of metallic systems.

A homogeneous intermediate-valence state can also be the basis for the formation of a long-range magnetic order in a rare-earth sublattice of Eu compounds, which is uncommon for most intermediate-valence systems. Using the example of the $EuCu_2(Si,Ge)_2$ system, the phase diagram and the essential features of this state are studied. A feasible ordering mechanism based on the possibility of the formation of an induced magnetic moment in a singlet-triplet system of Eu-ion spin-orbital states for one of the competing f-electron shell configurations is discussed.

From a more general point of view, in Sections 2 and 3, we presented and discussed physical phenomena whose mutual influence ensures the formation of one or another type of ground state for an f-electron strongly correlated system. Specifically, for the physics of induced magnetism (Section 2), these are the effects of the CEF and the interionic exchange interaction RKKY. An additional factor of the "back burner" here is hybridisation, which causes the effective "suppression" of local magnetic moments due to strong spin fluctuations exceeding the thermodynamic ($\sim k_B T$) limit.

For europium-based systems (Section 3), an important factor is already a competition of exchange interaction and hybridisation directly on the same rare-earth ion. Moreover, hybridisation causes a radical transformation of the spectrum of f-electrons up to the formation in $EuCu_2(Si_{1-x}Ge_x)_2$ ($x < 0.9$) spin gap, as in Kondo insulators. At the same time, however, the metallic nature of the conductivity remains, which is most likely a consequence of the multiplicity of types of electronic states near Fermi energy; not all of them participate in effective hybridisation with f-electrons



It is quite possible that it is hybridisation that underlies the renormalisation of the energy of the spin-orbit interaction (it determines the splitting of the J multiplet of the initial configuration $f^6$) for the IV ion Eu. The nature of this amazing phenomenon, which has also been observed for IV cerium ion systems [71,72], requires further and detailed analysis. However, just this renormalisation underlies the proposed approach to explaining the features of the magnetic phase diagram for the IV system $EuCu_2(Si,Ge)_2$ from the standpoint of the physics of induced magnetism.

The above considerations allow us to consider all three interactions, as noted in the introduction, in a close relationship and present them as shown in Figure 11.

This diagram illustrates the possibility of the formation of a number of different ground states as a result of the mutual influence of the three main interactions. The crystal field underlies the formation of f-electronic states in all crystalline rare-earth systems. Depending on the relative strength of the exchange interaction, one or another magnetically ordered state is formed. As the hybridisation increases, the effects associated with the conduction band start to dominate. This leads to a sequence of new states: a heavy-fermion state, a Kondo-insulator state, and, finally, an intermediate-valence spin-fluctuation state. Thus, the combination of effects of the three main interactions causes the appearance of phase transitions and a variety of types of ground states of the system.

The results presented in this review can be correlated with the CEF–RKKY line (for PrNi with substitution) and with the RKKY–Hybridisation line (for $EuCu_2(Si,Ge)_2$) in Figure 11. The results of many other experimental studies on the physics of SCES can be directly compared with this "triangle of interactions".

**Author Contributions:** Conceptualisation, P.S.S. and P.A.A.; methodology, P.S.S. and P.A.A.; supervision, P.A.A.; project administration, P.A.A.; writing—original draft preparation, P.S.S.; writing—review and editing, P.S.S. and P.A.A. All authors have read and agreed to the published version of the manuscript.

**Funding:** This work was supported by the Ministry of Science and Higher Education of the Russian Federation (Agreement No. 75-15-2021-1352).

**Acknowledgments:** The authors are deeply grateful to A. P. Menushenkov for the initiation of this work, and appreciative to A. V. Belushkin, A. V. Mirmelshtein, D. P. Kozlenko, and A. V. Mikheenkov for stimulating interest in the present study. We thank E. S. Clementyev and K. S. Nemkovsky for fruitful discussions on topics related to the subject of this work.

**Conflicts of Interest:** The authors declare no conflict of interest.

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
