# Peer review of "Uncommon Magnetism in Rare-Earth Intermetallic Compounds with Strong Electronic Correlations"

_crystals, doi:10.3390/cryst13081238_

Round 1

Reviewer 1 Report

Comments and Suggestions for Authors

This is a very nice review on rare-earth intermetallic compounds that exhibit magnetism caused by other means than RKKY interaction. The authors provide some theoretical background and then discuss several objects demonstrating magnetism because of various types of non-RKKY interaction. The review is well-written and structured nicely. Its layout reminds me of a PhD thesis, which high quality is obvious. If I were a board member, I would definitely vote for graduation. I eagerly recommend accepting it after only minor revision. Please see my comments below.

1.    Page 4. The description of the two-singlet model would benefit from a figure illustrating relative positions of the energy levels.

2.     Page 7. The description of the crystal structure of PrNi is a repetition. The same structure is already discussed on page 6.

3.    Page 8. If Gd ions have J = 0, then, in fact, they are Gd3+ ions with the f7 configuration.

4.    Page 10. “…in the 0.25Ce0.85Ni sample” is incorrect. Obviously, the symbol of Pr is missing in the formula; should be “Pr0.25Ce0.85Ni”.

5.    Figure 9. Why is the valence axis plotted from 2.1 to 2.9? It would be interesting to see the intersection of the dotted line on panel (c) with the energy axis at v equals exactly 2. If E(s-o) > 0 at v = 2, i.e., for J = 0, one might suspect that the extrapolation is incorrect, which would require further discussion.

6.    Conclusions are a bit confusing. They start with particular details pertinent to some half of the objects surveyed in the review and then evolve into a global picture of interactions in intermetallic RE compounds based only on a portion of the discussion. This looks imbalanced with respect to Sections 2 and 3. Could this section be improved?

Author Response

We thank the reviewers for their assessment of our work and their insightful comments.  They have helped us to improve the clarity of the manuscript. Below, we provide a point-by-point response to each of the Reviewers' questions and comments, and indicate the changes made in the revised text. With these changes, we believe that the manuscript is improved and now suitable for publication, and we hope that the Reviewers will support our opinion.

  1. Page 4. The description of the two-singlet model would benefit from a figure illustrating relative positions of the energy levels.

The new Figure 1, illustrating the ideas of the two-singlet model of induced magnetism, has been added to the manuscript.

  1. Page 7. The description of the crystal structure of PrNi is a repetition. The same structure is already discussed on page 6.

The repetition of the description of the crystal structure of PrNi in the text of the manuscript has been corrected.

  1. Page 8. If Gd ions have J = 0, then, in fact, they are Gd3+ ions with the f7 configuration.

Roughly saying this make sense, but the ion is still Eu due to the nuclear charge! Therefore the quantum numbers are identical but electron energy – not. For instance – similar situation is realized for Sm2+ and Eu3+ ions with f6-electron configuration. But energy of spin-orbit transitions for J=0 – J=1 configurations are different, that are 36 meV for Sm3+ and 45 meV for Eu3+.

  1. Page 10. “…in the 0.25Ce0.85Ni sample” is incorrect. Obviously, the symbol of Pr is missing in the formula; should be “Pr0.25Ce0.85Ni”.

The typo has been corrected

  1. Figure 9. Why is the valence axis plotted from 2.1 to 2.9? It would be interesting to see the intersection of the dotted line on panel (c) with the energy axis at v equals exactly 2. If E(s-o) > 0 at v = 2, i.e., for J = 0, one might suspect that the extrapolation is incorrect, which would require further discussion.

We selected the limits of the abscissa scale indicated in Fig. 10 (former Fig.9) basing on the maximum and minimum values of valence obtained during its measurements for the Eu-based set of the samples under discussion. A simple extrapolation of the available data does indeed allow us to expect the extrapolated value to coincide with the true value of 45 meV for the spin-orbit transition in Europium for valence 3+. For valence 2+, extrapolation gives a value close to 0 within the range of error-bars. It is noted in the discussion (Sec. 3.2) that such a result is the subject of a special separate theoretical analysis and in this paper is used only as an experimental fact.

A significant renormalization of the spin-orbit transition in the region of medium valence values (~ 2.45, i.e. far from the limit values) ~ is important for the formulation of a possible mechanism for the occurrence of a long–range magnetic order according to the scenario of induced magnetism. The latter is associated with a singlet-triplet scheme of f-electron levels and a large (about 7 barns for neutron cross-section) value of the matrix element connecting them. Just illustration of this is one of the main purposes of this drawing.

  1. Conclusions are a bit confusing. They start with particular details pertinent to some half of the objects surveyed in the review and then evolve into a global picture of interactions in intermetallic RE compounds based only on a portion of the discussion. This looks imbalanced with respect to Sections 2 and 3. Could this section be improved?

The conclusion has been significantly expanded and improved: the main results of the work are now more completely and specifically described in the context of the diagram of fundamental interactions.

Reviewer 2 Report

Comments and Suggestions for Authors

The authors Savchenkov et al, reported their work on titled, Uncommon magnetism in rare-earth intermetallic compounds with strong electronic correlations. Although, this work contains some results, the organization and interpretation of the result should be enhanced further. Hence, I recommend this work required a substantial revision before considering for publications.

1.      Provide the obtained results in the abstract in more concise.

2.      Highlights should be revised according to the journal standard.

3.      Novelty of the work should be highlighted in the introduction in more clearly.

4.      The authors designed the work nicely, merely presented the results but failed to discuss the observed results elaborately.

5.      I suggest the authors to compare the previous literature similar to that work to find a merits of this work.

6.      Refer and include the following references to strengthen the current version of the draft; Light-Science & Applications 11 (2022) 250; Current Opinion in Solid State & Materials Science 24 (2020) 100805.

Comments on the Quality of English Language

The authors Savchenkov et al, reported their work on titled, Uncommon magnetism in rare-earth intermetallic compounds with strong electronic correlations. Although, this work contains some results, the organization and interpretation of the result should be enhanced further. Hence, I recommend this work required a substantial revision before considering for publications.

1.      Provide the obtained results in the abstract in more concise.

2.      Highlights should be revised according to the journal standard.

3.      Novelty of the work should be highlighted in the introduction in more clearly.

4.      The authors designed the work nicely, merely presented the results but failed to discuss the observed results elaborately.

5.      I suggest the authors to compare the previous literature similar to that work to find a merits of this work.

6.      Refer and include the following references to strengthen the current version of the draft; Light-Science & Applications 11 (2022) 250; Current Opinion in Solid State & Materials Science 24 (2020) 100805.

Author Response

We thank the reviewer for his\her assessment of our work and his\her insightful comments.  The reviewer's comments have significantly contributed to the improvement of our manuscript. They have helped us to improve the clarity of the manuscript. Furthermore, we would like to extend a special note of thanks for the excellent examples of good practices that the reviewer provided in the comment #6. These articles represent excellent examples of well-executed works and have been taken into consideration during the improvement of our manuscript (Ref.8 in the new version of the manuscript).

Below are the main changes made in the revised text based on the reviewer's comments  #1-5:

  • (p.5) The new Figure 1, illustrating the ideas of the two-singlet model of induced magnetism, has been added to the manuscript. This will ensure a clearer understanding of the model by the reader.
  • (p.17) The conclusion has been significantly expanded and improved: the main results of the work are now more completely and specifically described in the context of the diagram of fundamental interactions.
  • (p.1) The abstract has been improved.
  • (p.15) A comparison between the results of the present work and the results of recent studies [1,2] has been added.

[1] Ahmida, Mahmoud A., et al. "Charge fluctuations across the pressure-induced quantum phase transition in EuCu2(Ge1− xSix)2Physical Review B 101.20 (2020): 205127.

[2] Ahmida, Mahmoud A., et al. "Interplay between valence fluctuations and lattice instabilities across the quantum phase transition in EuCu2(Ge1−xSix)2." Physical Review B 102.15 (2020): 155110.

  • The article's English language has been enhanced, and a number of typos have been corrected.

With these changes, we believe that the manuscript is now suitable for publication, and we hope that the Reviewers will support our opinion.

Round 2

Reviewer 2 Report

Comments and Suggestions for Authors

Revised version can be accepted